# A Proteomics Approach Identifies RREB1 as a Crucial Molecular Target of Imidazo–Pyrazole Treatment in SKMEL-28 Melanoma Cells

**DOI:** 10.3390/ijms25126760

**Published:** 2024-06-20

**Authors:** Erika Iervasi, Gabriela Coronel Vargas, Tiziana Bachetti, Kateryna Tkachenko, Andrea Spallarossa, Chiara Brullo, Camillo Rosano, Sonia Carta, Paola Barboro, Aldo Profumo, Marco Ponassi

**Affiliations:** 1IRCCS Ospedale Policlinico San Martino, Proteomics and Mass Spectrometry Unit, L.go. R. Benzi, 10, 16132 Genova, Italy; erika.iervasi@hsanmartino.it (E.I.); gabriela.coronel@hsanmartino.it (G.C.V.); kateryna.tkachenko@hsanmartino.it (K.T.); camillo.rosano@hsanmartino.it (C.R.); paola.barboro@hsanmartino.it (P.B.); aldo.profumo@hsanmartino.it (A.P.); 2Department of Pharmacy, Section of Medicinal Chemistry, University of Genoa, Viale Benedetto XV 3, 16132 Genova, Italy; andrea.spallarossa@unige.it (A.S.); chiara.brullo@unige.it (C.B.); 3IRCCS Ospedale Policlinico San Martino, Nuclear Medicine Unit, L.go. R. Benzi, 10, 16132 Genova, Italy; sonia.carta@hsanmartino.it

**Keywords:** cancer cell line, imidazo–pyrazole, differential proteomic analysis, gene ontology, cutaneous melanoma, RREB1

## Abstract

Cutaneous melanoma is the most dangerous and deadly form of human skin malignancy. Despite its rarity, it accounts for a staggering 80% of deaths attributed to cutaneous cancers overall. Moreover, its final stages often exhibit resistance to drug treatments, resulting in unfavorable outcomes. Hence, ensuring access to novel and improved chemotherapeutic agents is imperative for patients grappling with this severe ailment. Pyrazole and its fused systems derived thereof are heteroaromatic moieties widely employed in medicinal chemistry to develop effective drugs for various therapeutic areas, including inflammation, pain, oxidation, pathogens, depression, and fever. In a previous study, we described the biochemical properties of a newly synthesized group of imidazo–pyrazole compounds. In this paper, to improve our knowledge of the pharmacological properties of these molecules, we conduct a differential proteomic analysis on a human melanoma cell line treated with one of these imidazo–pyrazole derivatives. Our results detail the changes to the SKMEL-28 cell line proteome induced by 24, 48, and 72 h of **3e** imidazo–pyrazole treatment. Notably, we highlight the down-regulation of the Ras-responsive element binding protein 1 (RREB1), a member of the zinc finger transcription factors family involved in the tumorigenesis of melanoma. RREB1 is a downstream element of the MAPK pathway, and its activation is mediated by ERK1/2 through phosphorylation.

## 1. Introduction

Ever since their first synthesis at the end of the 19th century [1], pyrazole derivatives have gained increasing relevance in medicinal chemistry. Pyrazole is a five-membered-heteroaromatic ring characterized by two adjacent nitrogen atoms. These structural characteristics enable pyrazoles to react with both acid and basic moieties, allowing them to either donate or accept hydrogen residues. This versatility allows the pyrazole ring to be incorporated into a vast number of chemical compounds [2]. Nowadays, due to their pharmacological properties, pyrazole derivatives are broadly employed as drugs in the treatment of inflammation, pain, infections, obesity, cancer, and so forth [3,4].

Tumors are the second leading cause of death worldwide. Among all skin cancers, melanoma represents about 5% of new skin tumor diagnoses and shows up to 80% of mortality [5]. This malignancy originates from a benign melanocytic naevus, goes through an in situ melanoma, and eventually evolves into a metastatic neoplasm. The late metastatic form of the disease often develops resistance to medical treatments [6,7], underscoring the critical need to improve chemotherapeutic agents.

The environmental risk factors most involved in the development of the disease are ultraviolet (UV) radiation from sunlight. The UV-B spectrum is particularly associated with the development of melanomas with a high mutational load [8]. In response to UV-induced DNA damage, skin keratinocytes produce a melanocyte-stimulating hormone, which binds the melanocortin 1 receptor. This interaction drives melanocytes to produce and release melanin [6]. Paradoxically, in melanomas, the induction of melanin pigmentation through its toxic intermediates can cause attenuation of all treatments [9].

Recently, we characterized the antiproliferative and antioxidant properties of a series of imidazo–pyrazoles, a class of poorly investigated pyrazole derivatives. In particular, (*E*)-*N’*-(4-((4-chlorobenzyl)oxy)-3-methoxybenzylidene)-2-phenyl-2,3-dihydro-1*H*-imidazo[*1,2-b*]pyrazole-7-carbohydrazide, compound **3e** (Appendix A), showed relevant antiproliferative activity against the skin melanoma cell line SKMEL-28 (IC_50_ = 3.08 ± 0.10 μM). In contrast, it exhibited growth inhibition against the normal embryonic fibroblast cell line GM 6114, which is comparable to the approved anticancer drugs Cis-Pt [10].

In this study, to investigate the pharmacological properties and the mechanism of action of these derivatives, we performed a differential proteomic analysis at three different times (24 h, 48 h, and 72 h) on the cutaneous melanoma cell line SKMEL-28 treated or not with compound **3e**. The goal was to identify the list of differentially expressed proteins (DEPs) and investigate how deregulated transcription factors (TFs) influence pathways by examining gene ontology annotations for biological processes (GO-BP) that resulted in being significantly altered. Remarkably, from TFs present in our DEPs list, we can emphasize the downregulation of the Ras-responsive element binding protein 1 (RREB1), a reliable marker for diagnosis and prognosis prediction in melanoma [11].

## 2. Results

### 2.1. Proteomic Analysis

In this paper, we realized a differential proteomic study on the cutaneous melanoma cell line SKMEL-28. The cells were treated with DMSO, the **3e** solvent, or with the promising imidazo–pyrazole compound **3e**, using a 3.08 μM concentration corresponding to its IC50 value [10].

From a total of 4066 protein accessions that were identified, 4064 were associated with at least one Gene Symbol (see Protein List file on results at PRIDE repository PXD049299) (Appendix A). A group of significant DEPs was identified for each time interval as a result of **3e** vs. DMSO treatments (−1 ≤Log_2_ Fold Change (FC) ≥ 1, *p*-value ≤ 0.05). The differential analysis revealed varying numbers of deregulated proteins at each time point, as shown in Table 1.

### 2.2. Gene Set Enrichment Analysis of Differential Expressed Proteins

Gene ontology (GO) of all DEPs for each time was used for the examination of pathways using ShyniGO 0.80 [12]. The analysis has revealed a notable surge in RNA processing via phosphorylation at the 24 h mark, which persists at 48 h and 72 h. At 24 h, the identified processes include RNA processing, mRNA processing, RNA 3′-end processing, and mRNA 3′-end processing, indicative of significant alterations in gene expression regulation (Figure 1A). Additionally, the enrichment of protein phosphorylation reveals intricate signaling pathway modifications, potentially deregulating crucial cellular responses to the drug intervention. These findings emphasize the complex nature of cellular responses to treatment, indicating a concerted effort to regulate transcriptional and post-transcriptional events in the initial hours of treatment. At 48 h, the deregulation of autophagy, protein localization to organelle, and Golgi vesicle transport biological processes (Figure 1B) suggests a pivotal role for cellular degradation pathways in response to middle-term drug treatment, other than alterations in intracellular trafficking events. Furthermore, the observed protein localization to organelles suggests targeted cellular compartmentalization. By 72 h, deregulated processes include modifications in intracellular trafficking alongside the persistence of autophagy processes (Figure 1C).

The number of common and exclusive DEPs from the first three GO-BPs ranked on the base of the FDR values are shown in Appendix A.

DEPs were further analyzed by ShyniGO 0.80 in terms of possible targets of microRNA families (Appendix A); in particular, an enrichment of targets of the tumor suppressor mir-296-3p and of the pro-oncogenic mir-411, mir-208a, and mir-208b [13,14,15] were observed at 24 h, 48 h, and 72 h (Table 2).

Furthermore, as melanocytes have skin features, an epidermidis-specific network-based functional analysis of DEPs was performed using Functional Module Detection [16] to better contextualize deregulated pathways within the normal tissue where melanoma originates (Figure 2; Appendix A). Specifically, following a 24 h treatment, processes involved in the growth and development mechanism of melanocytes occur, such as the response to the transforming growth factor beta (TGF-β), cytokinesis, the metabolic process of acetyl-coA, migration of the epithelium (Figure 2A). It was verified that in the case of malignant melanoma, there may be overexpression of metabolic genes such as fatty acid synthase (FASN) and acetyl-coA carboxylase (ACC) [17,18]. After 48 h treatment, we observed hormonal involvement such as the regulation of cholesterol esterification; the negative regulation of the intracellular steroid hormone receptor signaling pathway; the intracellular steroid hormone receptor signaling pathway, in addition to the regulation of the epithelial–mesenchymal transition; and differentiation of mesenchymal cells (Figure 2B). After 72 h treatment, some mechanisms already highlighted at 24 h were enriched, such as the TGF-β receptor signaling pathway, wound healing, and positive regulation of epithelial cell migration (Figure 2C).

### 2.3. Deregulated Transcription Factors following **3e** Treatment

Deregulated TFs were identified among DEPs at each time point, utilizing the Human Transcription Factors database [19] as a primary resource. Specifically, 9 TFs were detected at 24 h, 12 at 48 h, and 9 at 72 h; the four TFs ZNF611, CUX1, TFAP2B, and RREB1 were identified as down-regulated at all three time points (Figure 3A,B), and thus they were considered for further evaluations.

The potential role of down-regulated TFs in melanoma following drug treatment was evaluated. First, their mRNA expression levels in melanoma were examined by consulting data from the Skin Cutaneous Melanoma (TCGA, PanCancer Atlas) collection of cBioPortal (https://www.cbioportal.org/ (accessed on 8 April 2024)), reporting mRNA expression data for each patient. As shown in Appendix A, ZNF611, CUX11, TFAP2B, and RREB1 exhibited dysregulation in 6%, 11%, 2%, and 15% of patients, respectively. Notably, the majority of these dysregulations manifested as up-regulation (indicated by red bars) rather than down-regulation (indicated by blue bars). In addition, patients’ outcome in terms of probability of overall survival (pOS) was analyzed; TFs analysis was consistently associated with a poorer prognosis compared to patients lacking mRNA deregulation in these genes (Appendix A). In addition, pOS analyses of TFs, individually performed, revealed that the most unfavorable prognosis was associated with up-regulation of RREB1, whereas patients exhibiting overexpression of ZNF611 and CUX1 demonstrated a better prognosis (Appendix A). Therefore, RREB1 was selected as an ideal candidate for validation as a possible therapeutic target in melanoma of compound **3e** for the following reasons: (i) it was the only DEP highly down-regulated at all three time points; (ii) *RREB1* up-regulation is associated with melanoma; (iii) mRNA *RREB1* is deregulated in the highest percentage of melanoma patients with respect to the other TFs (15% of alterations in patients); (iv) **3e**-mediated *RREB1* down-regulation is consistent with its pathogenetic overexpression in melanoma concerning TCGA and PanCancer Atlas repositories.

To validate the **3e** effect on RREB1 expression, we investigated *RREB1* mRNA levels after 24 h, 48 h, and 72 h treatments (Figure 3C). In particular, in agreement with RREB1 protein evaluation obtained by mass spectrometry, we observed *RREB1* mRNA reduction of 40% and 60% after 24 h and 48 h drug treatment, respectively. Conversely, while mass spectrometry suggested a down-regulation in REEB1 protein also after 72 h of **3e** treatment, mRNA did not confirm such an observation. The reason could be ascribed to the different sensitivity of the two techniques or an *RREB1* mRNA restoring at 72 h, which has not yet been followed by protein RREB1 renovation. Furthermore, to shed light on this discrepancy, we examined some protein families that usually participate in multidrug resistance (i.e., MDR, MRP, etc). Our results clearly showed that there is no evidence of differential protein expression between 48 h and 72 h, excluding the chance of drug resistance involvement.

### 2.4. In Silico Drug Repositioning in Melanoma Cells

In order to find other possible drugs acting on melanoma similarly to the **3e** compound, in addition to searching for further evidence of the **3e** compound’s effectiveness, a drug repositioning analysis was performed using Connectivity Map (CMap).

CMap is a platform designed to investigate drug modes of action and reposition data. It works by comparing genes from user-provided “genetic hit lists” to a large reference differential gene expression (DE) database. The output is a list of molecules ranked starting from the compound that produces the most similar gene signature to the most different one with respect to ours [20]. This bioinformatic tool allows for comparing the obtained gene expression profiles (GEPs) with GEPs driven in seven tumor cell lines treated with more than 6000 drugs in order to identify drug-driven gene signatures linked to our treatment. In particular, among cell lines of the CMap database, gene signatures obtained in melanoma A375 cells were considered to carry out the analysis. As shown in Figure 4, the five top-ranked drug families inducing high connection of gene signatures with **3e** were selected. The Venn diagram displays exclusive and common groups at 24 h, 48 h, and 72 h. In particular, the class of Fibroblast Growth Factor Receptor (FGFR) inhibitors is the only drug family shared by all-time treatments. It is consistent with the observation that FGFR can be considered a therapeutic target in melanoma [21].

In the upper part of the figure, the table represents the five top-ranked classes of drug obtained from CMap analysis of the differential expressed proteins (DEPs), results obtained from the comparison between DMSO-treated and **3e**-treated cells at 24 h, 48 h, and 72 h. In the lowest part, Venn analysis shows drug classes shared among different time points.

## 3. Discussion

Several studies have demonstrated how melanoma progression is influenced by genetic mutations and microenvironmental alterations, wherein deregulated proteins facilitate tumor invasion. Moreover, during carcinogenesis and tumor progression, melanomas can affect host homeostasis through the release of neurohormonal and immune mediators [22]. However, while essential mechanisms in melanoma development have been elucidated in recent decades, many of the biological, biochemical, and pathological processes underlying cutaneous melanoma evolution remain unclear, with insufficient data to resolve these uncertainties [8]. Additionally, a ‘definitive’ therapy for treating this neoplasm has yet to be achieved. To investigate certain features of the imidazo–pyrazole compound, known as **3e**, a differential proteomics approach utilizing the SKMEL-28 cell line treated with the compound was performed. Our proteomic analysis substantiates previous observations regarding the antiproliferative, antiaggregant, and antioxidant properties of imidazo–pyrazole **3e** [10]. Indeed, enriched GO analysis of significantly deregulated proteins in melanoma cells in vitro suggests that pathways involving growth factor beta receptor signaling pathway, membrane organization, epithelial cell migration, regulation of epithelial to mesenchymal transition, and endocytosis are profoundly affected by imidazo–pyrazole **3e** compound at different times (Figure 2). A differential proteomics approach has enabled us to gain a broader understanding of the cellular response to this imidazo–pyrazole class of compounds. More precisely, the transcription factor protein RREB1 is markedly downregulated following treatment with compound **3e** at all time points. In particular, we observed a down-regulation of *RREB1* mRNA at 24 and 48 h, likely responsible for the decrease in RREB1 protein. Although we cannot exclude a structural interaction between compound **3e** and RREB1, our results suggest its regulation at the expression level, which may serve as a potential explanation for the influence of imidazo–pyrazole **3e** compound on melanoma cells.

All treatments show a clear cell response to **3e** when analyzing DEPs’ general GO-BP as well as the Kyoto Encyclopedia of Genes and Genomes (KEEGs) pathways contextualized in the tissue. Indeed, at 24 h (Figure 1A and Figure 2A), the presence of regulation in RNA processing, protein phosphorylation, growth factor transforming beta (TGF-β) regulation, the metabolic process of acetyl-coA, mechanisms related to cytokinesis and epithelial cell migration can be found. Also, we can find different DEPs contributing to these biological processes. The more interesting ones, confirmed by the bibliography on melanoma, are as follows: USP15 [23], TIPIN [24], TMC5 [25], TYMP [26], USP19 [27], USP8 [28], and TFAP2B [29]. At 48 h (Figure 1B), autophagy was found to be the most important BP, which is a crucial component of the cellular adaptive stress response. Indeed, we can find DEPs linked to this process with a high differential FC as ARFRP1 [30], BMI1 [31], MECR [32], USP19, and TRIM32 [33]. The most significant GOs contextualized in the tissue from which melanomas arise (Figure 2B, M1, and M2) are connected to the regulation of chromosome organization, negative regulation, and remodeling of “very low density” lipoprotein particles. Those tissue-specific deregulations are appropriate according to bibliographic resources. In malignant melanomas, over-expression of metabolic genes such as fatty acid synthase (FASN) and acetyl-coA carboxylase (ACC) can be found [18]. ACC is the speed-limiting enzyme that converts acetyl-coA to malonyl-coA, while FASN generates long-chain fatty acids from both acetyl-coA and malonyl-coA [34].

Other coherent results from our analysis of biological and molecular processes strongly involved in melanoma are RNA processing, phosphorylation mechanisms, lysosomal system involvement [35], cellular response to growth factor beta receptor signaling, and epithelial cell migration. Interestingly, it was observed that FGF-2 can promote angiogenesis by acting together with VEGF since FGF is a key regulator for the development of blood and lymphatic vessels, as it modulates endothelial metabolism driven by MYC-dependent glycolysis. In tumors, FGF expression has been associated with resistance to anti-angiogenic therapy. In fact, it has been hypothesized that the activation of the pro-angiogenic FGF signaling pathway represents a mechanism used by tumor cells to escape therapies that target VEGF [36].

Mitochondrial activity is strongly involved in melanoma processes, mainly in the production and regulation of reactive oxygen species (ROS). The precise mechanism of the elevated ROS could be related to the production of melanin itself since the same phenomenon has also been noted in benign melanocytes [37]. Low to moderate levels of ROS have been shown to promote growth and suppress apoptotic pathways, while high levels can stimulate apoptosis by activating pro-apoptotic factors such as Bax. ROS-responsive proteins play a role in the homeostasis and regulation of stress responses, with autophagy acting as a critical arm of the cell’s adaptive stress response [38].

Alternative splicing plays an important role in the effectiveness of immunotherapy, targeted therapy, and melanoma metastases. Abnormal expression of splicing factors and variants can serve as biomarkers or therapeutic targets for the diagnosis and prognosis of melanoma. Moreover, changes in lysosomal functions can contribute to various biological processes like inflammation and programmed cell death (apoptosis), further impacting melanoma development and progression [39]. Several studies have shown that lysosomal enzymes play a key role in invasiveness, angiogenesis, and cancer progression. Cancer cells must compete with continuous changes in the tumor microenvironment, which is poor in vascularization, with low levels of nutrients and oxygen: for this reason, the activity of lysosomes helps to generate those nutrients and the energy required by cancer cells to survive [40]. All these processes can be observed through the enrichment analysis of deregulated proteins by **3e**.

The frequent amplification of RREB1 in melanoma suggests its important role in melanoma tumorigenesis [41], and it is used as a good marker for tumor diagnosis, prognosis prediction, and patient management. Accurate classification of melanocytic tumors as benign or malignant is crucial for diagnosis and treatment. Histopathological detection is the standard method for melanoma diagnosis. However, distinguishing tumors with features of both nevi and melanoma is challenging. RREB1’s role in diagnosis is significant. Staining 6p25 (RREB1) through fluorescence in situ hybridization emerges as a highly sensitive diagnostic tool for melanoma, showing robust correlation across all cases and subtypes [42]. In this light, RREB1 downregulation could be considered a molecular target in melanoma.

The analysis with CMap [20] allowed us to confirm our assumptions about the action of compound **3e**, as a list was created with about 150 top-ranked up and down DEPs vs. line A375 (melanoma cell line) to find a list of drugs that showed similar feature and mechanisms of action to our compound. Only those classes of drugs with >95% connection were selected. Finally, for each time, the top-ranked drug classes were identified (Figure 5): the Venn diagram shows an element common to the three times, the Fibroblast Growth Factor Receptor (FGFR) inhibitor. FGFR, acting in combination with VEGF, promotes the mechanism of angiogenesis since FGF is a key regulator for the development of blood vessels and modulates endothelial metabolism [36].

### RREB1 Interactions with Significantly Deregulated Proteins

Recent studies have shown that RREB1 appears to be involved in biological processes such as DNA damage repair, cell growth and proliferation, cell differentiation, fat development, fasting glucose balance, zinc transport, and transcriptional regulation. As previously explained, an imbalance of RREB1 function plays a role in the development of various cancers and other diseases. Using DEPs exhibiting a-2 ≤ Log_2_ FC ≥ 2, we performed a reconstruction of RREB1 interactions with its closest neighbors (Figure 5A). Also, through STRING enrichment on Cytoscape, we predicted the molecular pathways existing between RREB1 and its closest DEP neighbors (Figure 5B). The algorithm predicted a direct interaction of RREB1 with KAT2B and TP53 genes; however, neither KAT2B nor TP53 proteins were identified by mass spectrometry in our SKMEL-28 model.

The absence of p53 among DEPs was expected, given previous observations of mutant *TP53* in SKMEL-28 [43], suggesting that the failure to regulate apoptosis processes can be ascribed to mutant p53 protein rather than down-regulation of p53 expression. Nevertheless, it has been demonstrated that RREB1 transactivates p53 expression, playing a pivotal role in p53-induced apoptosis in response to DNA damage [44]. In the bibliography, we found studies about our RREB1’s nearest neighbors, DEPs, predicted as TP53 interactors in Figure 5B, involved in apoptosis: PPP2R5C (also called B56γ) reduces p53 stabilization when ablated [45]; PPP4C is also implicated in DNA repair and cell cycle regulation [46]; USP15 has a role in the crosstalk between TGF-β signaling and p53 stability [47,48]; and WAPL is involved in checkpoint-dependent mitotic arrest [49].

Interestingly, there is an important regulation of clathrin-dependent endocytosis in Figure 2C module 1 at 72 h treatment, as well as receptor-mediated endocytosis. Both processes are detected by GIANT analysis in the epidermis to be linked to TUSC3, UBE2H, WAPL, and ZC3H7A deregulation. Clathrin-dependent endocytosis is an essential mechanism for internalizing ligand–receptor complexes that signal proliferation (EGF, insulin, IGF1), apoptosis (TNFα, TRAIL, Fas-L), differentiation, and morphogenesis (TGFβ, WNT, Notch, SHH) [50]. This process seems to be also linked to RREB1 through TP53-USP15-WAPL predicted regulation, as present in Figure 5B, but no previous research about WAPL linked directly to endocytosis has been reported. Instead, the role of WAPL in centromere cohesion and checkpoint-dependent mitotic arrest [51] and the role of USP15 in deubiquitinating TGF-βR1 [52] are well known, with both processes connected to other terms in 72 h modules, as shown in Figure 2C.

In conclusion, **3e** seems to deregulate pathways involved in melanoma pathogenesis, and RREB1 down-regulation could be one of the players in such an effect.

## 4. Materials and Methods

### 4.1. Cell Culture and Treatment

SKMEL-28 cell line (skin melanoma, Biologic Bank and Cell Factory, IRCCS Policlinico San Martino, Genova, Italy) was grown in DMEM (10% FBS, 2 mM Glutamine and 1% penstrep) and incubated at 37 °C in 5% CO_2_ in a humidified environment. All reagents were purchased from EuroClone (Milan, Italy). Cultured cells were seeded and treated in 75 cm^2^ flasks with compound **3e** (at conc. 3.08 μM, corresponding to its IC_50_ concentration) for 24 h, 48 h, and 72 h. Compound **3e** was freshly prepared and purified, as previously reported [10]. The same procedure was carried out with an equal volume of DMSO but with no compound, and these resulting DMSO-treated cells were used as a reference. When the treatment was completed, cells were washed twice with ice-cold 1×PBS and then harvested using cell scrapers. Centrifuged cell pellets were resuspended with RIPA buffer (completed with 1 mM DTT and protease and phosphatase inhibitors) and frozen at −80 °C.

### 4.2. Sample Preparation and Mass Spectrometry Analysis

All procedures were performed on ice or at 4 °C, except where specified. Once thawed, cell lysates were vortexed every 15 min four times and then sonicated for 30 s with pulse [at approximately 10 watts output]. Samples were centrifugated at 13,850 rcf for 10 min. Then, supernatants were collected, their volumes were measured, and the same amount of 20% SDS-6% DTT was added. Subsequently, they were incubated at 95 °C for 5 min. Once cooled, five volumes of MATF (Methanol, Acetone, and Tributhyl phosphate, 1:12:1) were added, and samples were incubated for 1 h with agitation. The total proteins were finally pelleted by centrifugation at 12,000 rcf for 15 min, the supernatants were removed, and the protein precipitates dried for about 30 min at RT in a Savant SpeedVac apparatus (Thermo Fisher Scientific, Waltham, MA, USA). The dried pellets were resuspended in 250 μL of 5% SDS in 50 mM Ammonium Bicarbonate [AMBIC]. Then, using the QuantiPro BCA Assay Kit (Sigma-Aldrich, St. Louis, MO, USA), each sample was processed to determine its protein concentration. Later, 20 mM DTT was added to 50 μg of each total protein extract, and the samples were incubated at 95 °C for 10 min. Then, when cooled to RT, Iodoacetamide was added to a final concentration of 40 mM, and the reaction mixtures were incubated for a further 30 min in the dark. Finally, orthophosphoric acid was added to a final 1.2% concentration. After reduction and alkylation, the acidified protein samples were loaded onto S-Trap mini columns, washed four times, and successively trypsin-digested as suggested by the manufacturer (Protifi, Farmingdale, NY, USA). The eluted mixtures were dried under vacuum at RT in a Savant SpeedVac apparatus (Thermo Fisher Scientific, Waltham, MA, USA).

To perform mass spectrometry analysis, two biological replicates of the same condition were generated, and they were all analyzed twice by LC/MS-MS to obtain a dataset with four observations per experimental point. Before loading samples, the desiccated tryptic digests were resuspended with 0.2% formic acid in water. Mass spectrometry analysis was performed by nano-UHPLC-MS/MS using an Ultimate 3000 chromatography system equipped with a PepMap RSLC C18 EASY spray column [75 μm × 50 cm, 2 μm particle size] (Thermo Fisher Scientific, Waltham, MA, USA) at a flow rate of 250 nL/min with a temperature of 60^◦^C. The following mobile phase composition was used: (A) 0.1% *v*/*v* formic acid in water; (B) 80% ACN, 20% H_2_O, and 0.08% *v*/*v* formic acid. A 105 min gradient was selected: 0.0–3.0 min isocratic 2% B; 3.0–7.0 min 7% B; 7.0–65.0 min 30% B; 65.0–78.0 min 45% B; 78.0–83.0 min 80% B; 83.0–85.0 isocratic 80% B; 85.0–85.1 2% B; and finally 85.1–105.0 isocratic 2% B. After separation, the flow was directly sent to an Easyspray source connected to a Q Exactive™ Plus Hybrid Quadrupole-Orbitrap ™ mass spectrometer (Thermo Fisher Scientific, Waltham, MA, USA). The data were acquired in data-dependent mode, alternating between MS and MS/MS scans. The software Xcalibur (version 4.1, Thermo Fisher Scientific, Waltham, MA, USA) was used to operate the UHPLC/HR-MS. MS scans were acquired at a resolution of 70,000 between 200 and 2000 *m*/*z*, with an automatic gain control [AGC] target of 3.0 × 106 and a maximum injection time [maxIT] of 100 ms. MS/MS spectra were acquired at a resolution of 17,500 with an AGC target of 1.0 × 105 and a maxIT of 50 ms. A quadrupole isolation window of 2.0 *m*/*z* was used, and HCD was performed using 30 normalized collision energy [NCE].

### 4.3. Protein Identification and Data Analysis

The mass spectrometry *.RAW data and identification files were deposited into the ProteomeXchange Consortium via the PRIDE [53] partner repository with the dataset identifier PXD049299. All *.RAW format files were processed with ProteomeDiscoverer^®^ software version 2.4.1.15 (Thermo Fisher Scientific, Waltham, MA, USA) for PMS identification, protein quantification, and differential proteomic analysis. Briefly, in the processing step, the following were established: the database for PMSs identification in MS/MS spectra and concatenated decoy (Homo sapiens—sp_canonical v2023-06-28; Target FDR strict = 0.01; Target FDR relaxed = 0.05 for proteins, peptides, and PSMs), setting static modification (Carbamidomethyl/+57.021Da on C) and dynamic modifications (oxidation/+15.995 Da (M); phospho/+79.966 Da (S, T, Y)), and tolerances (precursor mass tolerance = 10 ppm, fragment mass tolerance = 0.02 Da) for the used identification engines (MS Amanda 2.0, Sequest HT^®^) [54,55]. In the consensus step, precursor abundance was calculated by intensity, using Unique+Razor peptides and considering proteins for peptide uniqueness. Peptide normalization (based on total peptide amount, scaling on all averages), peptide filters (high confidence, minimum length = 6), protein quantification (by peptide’s summed abundances), and differential expression (pairwise ratio and t-test background based) were also assessed in this step using IMP-apQuant node.

### 4.4. Pathways Analysis

DEPs were filtered by applying the following criteria: Log_2_FC ≥ 1 U Log_2_FC ≤ −1, with *p*-value < 0.05. DEPs list underwent gene-set enrichment analysis by using ShinyGO 0.80 (http://bioinformatics.sdstate.edu/go80/ (accessed on 8 March 2024)). In particular, GO Biological Processes (GO-BP) with a false discovery rate (FDR) < 0.05 were considered. Fold enrichment, defined as the percentage of genes in your list belonging to a pathway divided by the corresponding percentage in the background, number of genes, and FDR, was evaluated to rank the most significant GO-BP. A pathway redundancy filter was also applied. For microRNA family prediction, the ShinyGO miRNA.target.microRNA was used. To better contextualize deregulated pathways, HumanBase’s Functional Module Detection tool [16] (https://hb.flatironinstitute.org/ (accessed on 8 March 2024)) was used. This tool finds cohesive gene clusters from a provided gene list selecting relevant tissue, in this case, “epidermidis”.

### 4.5. Connectivity Map

Connectivity Map (CMAP) Connectopedia (https://clue.io/connectopedia/ (accessed on 8 April 2024)) was used to identify drug-driven gene signatures connected to **3e** treatment. In particular, among the seven available cell lines, melanoma A375 was investigated for connected gene expression signatures following input of the 100 (or 150 if available) top-ranked-up and -down DEPs obtained by MS analysis after 24 h, 48 h, 72 h of **3e** treatments. Drugs were grouped into functional classes for each time. Venn analysis (https://www.interactivenn.net/ (accessed on 8 April 2024)) was performed to identify common classes along time points.

### 4.6. RNA Extraction and Real-Time PCR

SKMEL-28 cells were plated in 60 mm dishes and added with **3e** or DMSO; after 24 h, 48 h, and 72 h, RNA was extracted with RNEasy Plus Mini Kit (Qiagen); RNA quality was assayed by NanoDrop 1000 spectrophotometer (Thermo Scientific). cDNA was obtained starting from 1 μg RNA with iScript cDNA Synthesis kit (BioRad) following the manufacturer’s instructions. Real-time PCR was performed for RREB1 expression with the following Taqman assays (Life Technologies): target RREB1 Hs00366107_m1; housekeeping GAPDH Hs02786624_g1; and beta-2-microglobulin 4331182 Hs00187842_m1. Reactions were run on Mastercycler^®^ ep realplex (Eppendorf), and an analysis was performed using Realplex software(Version 2.5.2).

## 5. Conclusions

In this paper, by conducting a differential proteomics analysis, we described how an imidazo–pyrazole treatment at various time points can modify the proteomic expression of a cutaneous melanoma cell line. The findings and insights generated from this study support the importance of RREB1 and its role in melanoma tumorigenesis by inhibiting the action of tumor suppressor genes. Furthermore, even though our study was realized using only the SKMEL-28 cell line and that confirmation with other melanoma cell lines could be essential, all these results could potentially represent a valuable contribution towards the development of novel and improved chemotherapeutic agents for patients struggling with this serious illness.

## Figures and Tables

**Figure 1 ijms-25-06760-f001:**
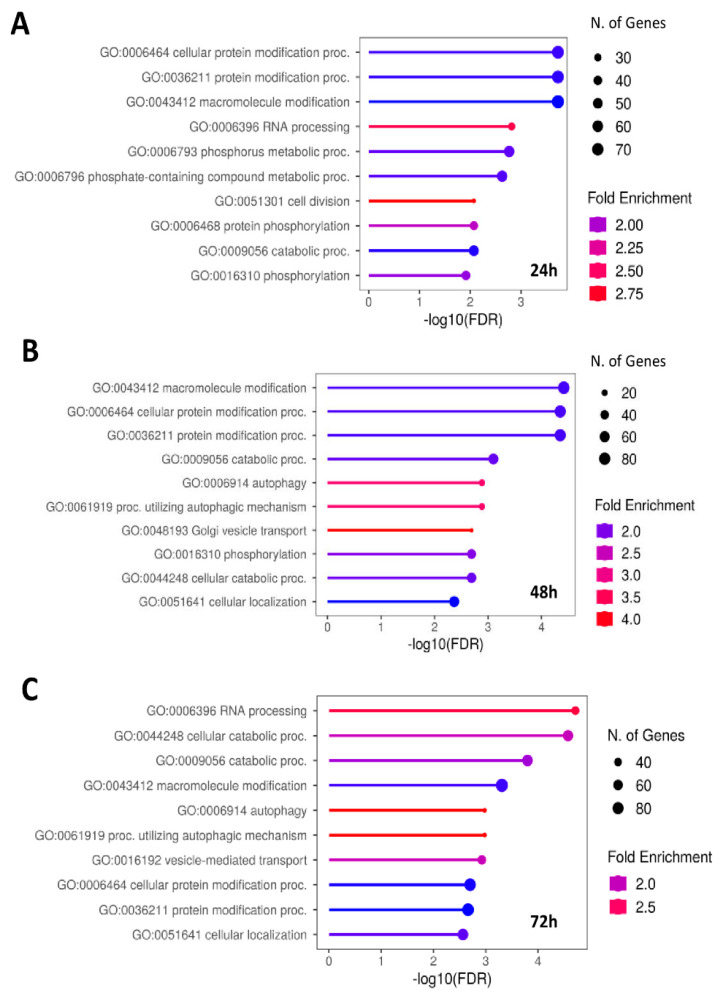
Pathways analysis in SKMEL-28 cells treated with compound **3e**. Representation of gene set enrichment analysis of differentially expressed proteins (DEPs) obtained from the comparison between DMSO-treated and **3e**-treated cells for 24 h (**A**), 48 h (**B**), and 72 h (**C**). Gene Ontology Biological Processes (GO-BP) from ShyniGO 0.80 are represented in lollipop diagrams, ranked based on false discovery rate (FDR) values (longer bar corresponds to lower FDR), where fold enrichment is shown by color-scale from blue (lowest) to red (highest) and the number of DEPs is indicated by circles diameters (from 20 to 80).

**Figure 2 ijms-25-06760-f002:**
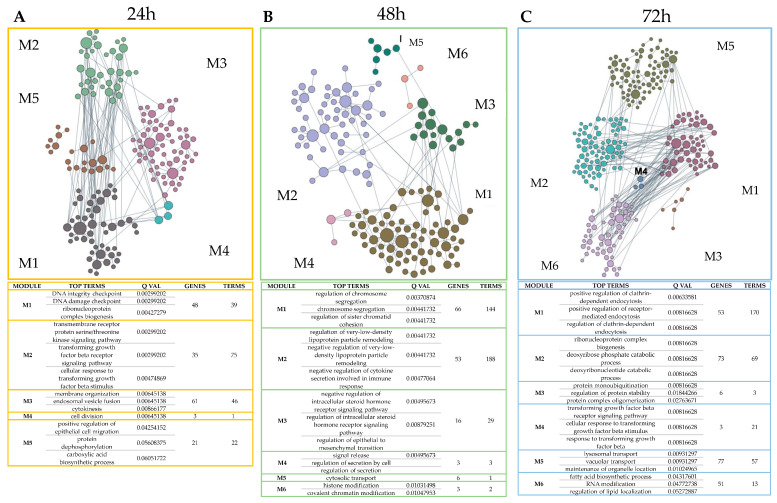
Genome-wide Integrated Analysis of gene Networks (GIANT) in epidermis using DEPs of SKMEL-28 cells treated with **3e**. Pathways deregulated in the healthy epidermis context are obtained from analysis of differentially expressed proteins (DEPs) acquired from comparison between DMSO-treated and **3e**-treated cells for 24 h (**A**), 48 h (**B**), and 72 h (**C**), performed by HumanBase Functional Module Detection tool. Modules (M) represent the groups of deregulated pathways and are numbered starting from the highest to the lowest globally statistically significant. For each module, the top three pathways are shown and ranked as above.

**Figure 3 ijms-25-06760-f003:**
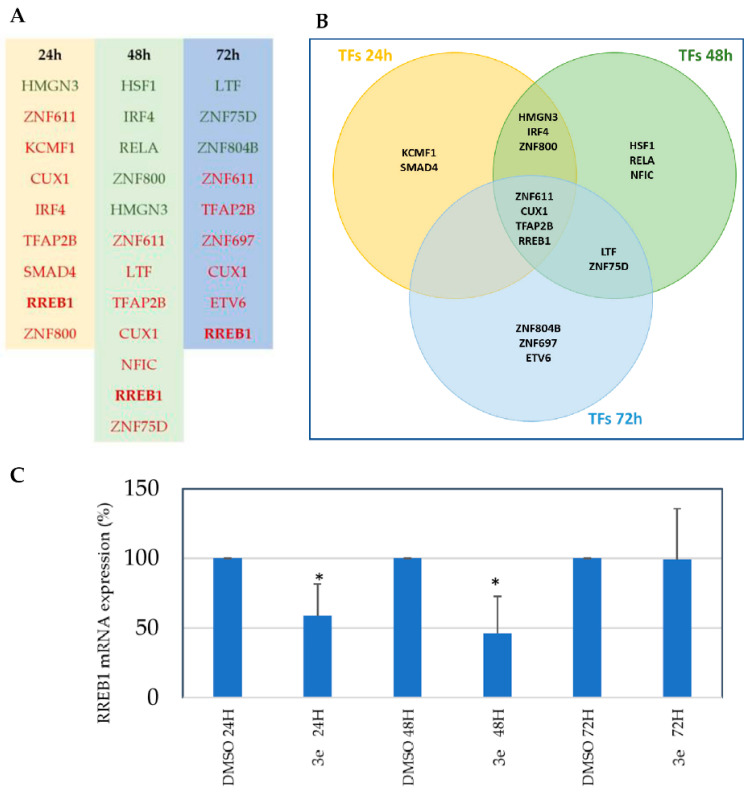
Contribution of transcription factors in **3e** mediated SKMEL-28 proteome deregulation. (**A**) The list of transcription factors (TFs) among the differential expressed proteins (DEPs), obtained from a comparison between DMSO-treated and **3e**-treated cells for 24 h (9 TFs), 48 h (12 TFs), and 72 h (9 TFs), is shown. Up-regulated TFs are in green; down-regulated TFs are in red. (**B**) Venn analysis represents common deregulated TFs shared by different time points. (**C**) Analysis of RREB1 mRNA expression performed on DMSO-treated and **3e**-treated cells for 24 h, 48 h, and 72 h. Results are the mean ± SD of three independent experiments performed on three different cDNA preparations. GAPDH and beta-2-microglobulin were used as housekeeping genes to normalize results. The asterisk indicates a statistically significant difference between DMSO- and **3e**-treated cells at 24 h and 48 h time points.

**Figure 4 ijms-25-06760-f004:**
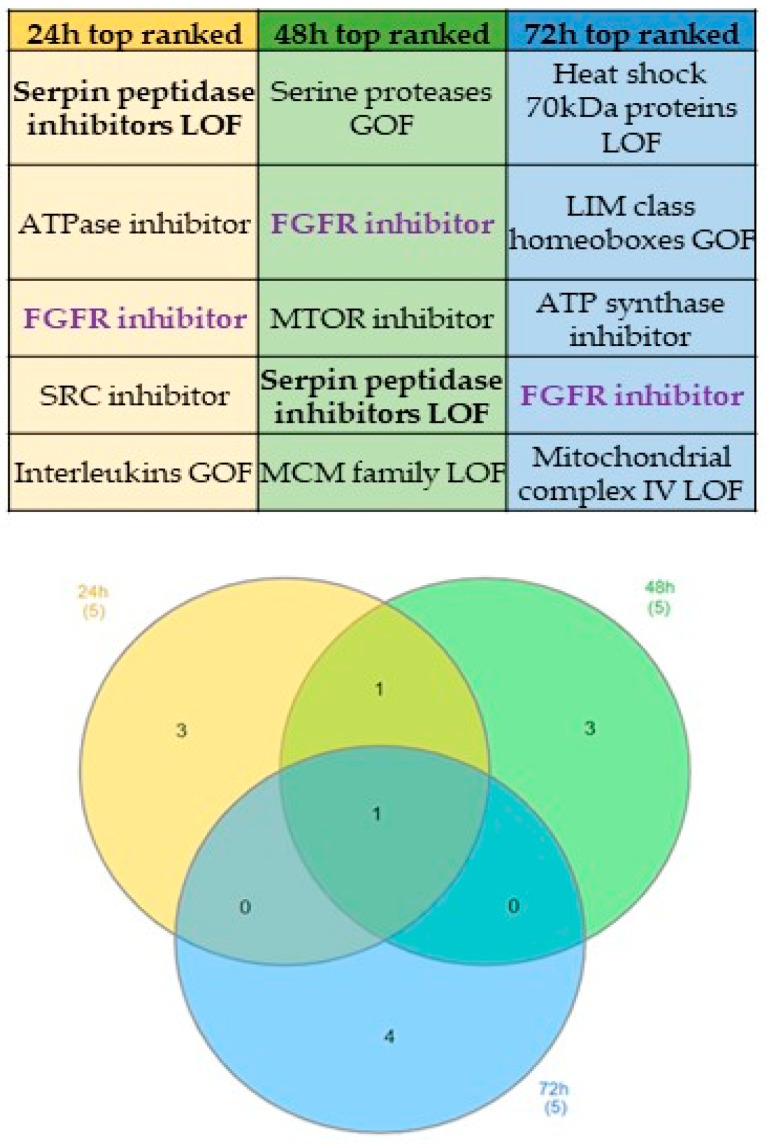
In silico analysis of **3e** connected drugs in melanoma cells.

**Figure 5 ijms-25-06760-f005:**
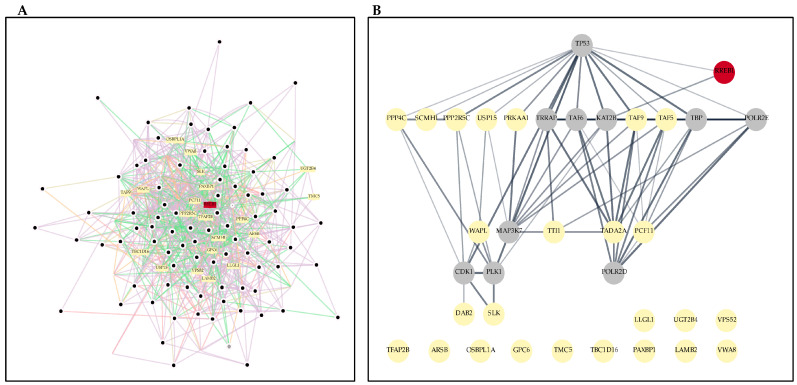
RREB1 protein–protein interactions network: (**A**) Cytoscape’s GeneMANIA Protein–Protein Interaction (PPI) Network of Differentially Expressed Proteins (DEPs) with Fold Change (FC) ≥ 1, in red RREB1, yellow nodes are first neighbors of RREB1: ARSB, DAB2, GPC6, LAMB2, LLGL1, OSBPL1A, PAXBP1, PCF11, PPP2R5C, PPP4C, PRKAA1, SCMH1, SLK, TADA2A, TAF5, TAF9, TBC1D16, TFAP2B, TMC5, TTI1, UGT2B4, USP15, VPS52, VWA8, and WAPL. (**B**) STRING prediction of molecular pathways using RREB1 interactions with its closest DEP neighbors; red—RREB1; yellow—proteins identified in our study; grey—predicted missing proteins.

**Table 1 ijms-25-06760-t001:** Numbers of DEPs at 24 h, 48 h, and 72 h of **3e** treatment.

DEPs 24 h	DEPs 48 h	DEPs 72 h
273	272	355

**Table 2 ijms-25-06760-t002:** List of micro-RNA families at 24 h, 48 h, and 72 h.

miRNA 24 h	miRNA 48 h	miRNA 72 h
Hsa-miR-219-5p	Hsa-miR-339-5p	Hsa-miR-486-5p
Hsa-miR-486-5p	Hsa-miR-411	Hsa-miR-219-5p
Hsa-miR-208a/208b	Hsa-miR-208a/208b	Hsa-miR-133a/133b
Hsa-miR-296-3p	Hsa-miR-142-3p	Hsa-miR-202
Hsa-miR-411	Hsa-miR-125-5p	Hsa-miR-339-5p
	Hsa-miR-296-3p	Hsa-miR-296-3p
	Hsa-miR-125b	Hsa-miR-451
	Hsa-miR-196b	Hsa-miR-411
		Hsa-miR-192
		Hsa-miR-215
		Hsa-miR-196b
		Hsa-miR-208a/b
		Hsa-miR-126
		Hsa-miR-615-3p
		Hsa-miR-875-5p
		Hsa-miR-142-3p
		Hsa-miR-125a-5p

## Data Availability

Data are contained within the article and Appendix A.

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
