# Peer review of "A Proteomics Approach Identifies RREB1 as a Crucial Molecular Target of Imidazo–Pyrazole Treatment in SKMEL-28 Melanoma Cells"

_ijms, 2024, doi:10.3390/ijms25126760_

Round 1
Reviewer 1 Report
Comments and Suggestions for Authors
Comments attached

Author Response
Response to Reviewer 1
Thank you so much for the helpful comments to our manuscript.
- In the result section please describe the concentration and its optimization of 3e compound treated to the SKMEL-28 cell line.
We added the sentence “In this paper, we realized a differential proteomic study on the cutaneous melanoma cell line SKMEL-28. The cells were treated with DMSO, the 3e solvent, or with the promising imidazo-pyrazole compound 3e, using a 3.08 mM concentration corresponding to its IC50 value [9].” See lines 84-87.
- The numbers of differential proteins expression shown in table S1, should be provided in the main text of the manuscript in the form of graphical presentation in figure format.
We added table 1 in the main text. See lines 97-99.
- In figure1 the author described the DEP which are indicated by number 10-40 in circle representing the number of Genes, here the author group need to provide the common and specific DEP/genes in time points 24 hrs., 48 hrs. and 72 hr. treatment.
We added the sentence “The number of common and exclusive DEPs from the first three GO-BP, ranked on the base of the FDR values, are shown in Fig.S2.” (see lines152-153) and we also added Fig. S2 in the supplementary materials together with Table S2 for the list of genes we have used to build the Venn diagram.
- The data of micro-RNA families described in Figure S2 should be in the main text.
We added Table 2 in the main text, see lines 157-159
- In figure 3B, the van analysis presentation, the down regulated number count of TFs has been mentioned in the circles, instead of the number the names of the TFs should be place in their respective position, for example all three-time point detected four TFs ZNF611, CUX1, TFAP2B and RREB1 should be place in the circle position at all three-point position.
We changed numbers with the name of the Venn diagram of TFs in Fig. 3B, see lines 194-195
- In figure 3C, the author stated that mRNA analysis of RREB1 expression not supporting the mass spectrophotometry data by not showing the significant down regulation pattern at 72-hr treatment, at this point the quantification of the mass spectrophotometry data should be carried out or there should be additional treatment with e3 compound for 96 hrs to establish the proof of principle of the down regulation of RREB1 by the treatment of e3 compound.
A differential proteomic analysis performed through data dependent (DDA) Mass Spectrometry doesn’t allow any “absolute” concentration/quantification as result. In the table S1 (DEPs by time), you can find the “relative” abundance of DEPs. Moreover, to set a 96h 3e treatment and use it to perform a further time point proteomics analysis, will be technically difficult due to the high number of dead cells already present at 72h which could not allow to obtain reliable results.
- To address the issue of RREB1’s non-significant down regulation pattern at 72-hrs treatment, the author should explore the potential drug resistance in SKMEL-28 cell line treated by e3 compound.
To address this observation, we added the sentence “Furthermore, to shed light on this discrepancy, we examined some protein families usually participating in multidrug resistance (i.e. MDR, MRP etc). Our results clearly showed that there is no evidence of differential protein expression between 48h and 72h (data not shown), excluding the chance of a drug resistance involvement” as you can see in lines 234-238.
Reviewer 2 Report
Comments and Suggestions for Authors
Dear authors, greetings for your manuscript.
ABSTRACT : I suggest to rewrite more fluently some phrases. In particular I suggest to use "cutaneous melanoma " in line 1 and 2 .
About RREB1 you have to write about its dependence on MAPK pathway (as it is described in reference 10 that you will cite in the text at the end of introduction chapter).
RESULTS:
FIGURE 1 : I think it could be necessary add data about SKMEL-28 UNTREATED (dmso at time 0) . How can readers see dmso treated level? Can you add graphical tool for it?
I suggest to add reference in line 2 page 4 about miRNA as tumor suppressor and their target .
When you write "melanocytes have skin features" and you analyse deps in epidermis, you refere to normal tissue. So I suggest to specify it in the text and in the figure 2 legend.
Figura 3 : statistically significant in 72H 3e treated is not observable.
CONCLUSIONS:
I suggest to add that your data confirm RREB1 importance and role in cutaneous melanoma as cited previously (ref. 10) .
Good luck
Thank you
Comments on the Quality of English LanguageDear authors
I suggest to rewrite clearer the first part of abstract.
Author Response
Response to Reviewer 2
Many thanks for your useful comments to our manuscript.
ABSTRACT:
- I suggest to rewrite more fluently some phrases. In particular I suggest to use "cutaneous melanoma " in line 1 and 2.
We added “cutaneous”in the Abstract section (see lines 22 and 23), in the Keywords (line 39), in the Introduction (line 72), in the Discussion (line 282) and in the Conclusion (line 546).
- About RREB1 you have to write about its dependence on MAPK pathway (as it is described in reference 10 that you will cite in the text at the end of introduction chapter).
We included the sentence “RREB1 is a downstream element of the MAPK pathway and its activation is mediated by ERK1/2 through phosphorylation” (see lines 36-37).
RESULTS:
- FIGURE 1: I think it could be necessary add data about SKMEL-28 UNTREATED (dmso at time 0). How can readers see dmso treated level? Can you add graphical tool for it?
This study is a differential proteomic analysis and we have not inserted “not-DMSO treated” cells. In fact, the data set obtained from DMSO treatments have been used to “normalize” the 3e effects on its solvent DMSO. In this way, the final DEPs dataset is the result only of compound 3e.
- I suggest to add reference in line 2 page 4 about miRNA as tumor suppressor and their target.
We add ref. [12-14] as you can see in line 156.
- When you write "melanocytes have skin features" and you analyse deps in epidermis, you refere to normal tissue. So I suggest to specify it in the text and in the figure 2 legend.
We add “normal” in line 163 and “healthy” in the figure 2 legend (line 181).
- Figura 3: statistically significant in 72H 3e treated is not observable.
We remove “at each time points” in line 206 and we add “at 24h and 48h time points”.
CONCLUSIONS:
- I suggest to add that your data confirm RREB1 importance and role in cutaneous melanoma as cited previously (ref. 10).
We have address this point with the sentence “The findings and insights generated from this study support the importance of RREB1 and its role in melanoma tumorigenesis by inhibiting the action of tumour suppressor genes.” (see lines 546-548).
- I suggest to rewrite clearer the first part of abstract.
We have made other changes not only in the abstract but in all the manuscript’s sections.
Round 2
Reviewer 2 Report
Comments and Suggestions for Authors
Dear authors thank you to added revisions for submission.
Author Response
Many thanks for your helpful comments to our manuscript. They really improved our job.